# Microstructures and Mechanical Properties of SUS 630 Stainless Steel: Effects of Age Hardening in a Tin Bath and Atmospheric Environments

**DOI:** 10.3390/ma18030574

**Published:** 2025-01-27

**Authors:** Kuan-Jen Chen, Fu-Sung Chuang

**Affiliations:** Department of Mechanical Engineering, Southern Taiwan University of Science and Technology, Tainan 710, Taiwan; mb110111@stust.edu.com

**Keywords:** precipitation hardening, 630 stainless steel, aging media, Sn bath, low carbon emissions

## Abstract

This study investigates the solution-aging treatment of precipitation-hardening SUS 630 stainless steel, alongside an analysis of the carbon emissions generated by the energy consumed during aging treatments. By employing atmospheric and liquid tin as aging media, the research comprehensively explores the effects of aging treatments on the characteristics of 630 stainless steel. The maximum hardness value for the 630 stainless steel was observed after atmospheric aging at 500 °C for 1 h. The given 630 stainless steel obtained its maximum hardness value after atmospheric aging at 500 °C for 1 h, indicating that the formation of secondary precipitates strengthens the steel’s performance. By leveraging the intrinsic characteristics of liquid tin, using it as an aging medium (Sn bath aging) significantly improves the efficiency of the aging process, achieving mechanical properties comparable to those of atmosphere-aged steel. The 630 stainless steel aged in a Sn bath exhibited a refined martensitic matrix with substantial precipitate formation, contributing to superior impact toughness and dynamic fatigue resistance. With an equivalent mass and performance, Sn bath aging notably reduced the duration of the treatment compared to atmospheric aging, leading to substantial energy savings and reduced carbon emissions. The Sn bath treatment, recognized in metallurgical science and heat treatment for its excellent thermal conductivity and recyclability, shows potential to enhance process efficiency and enable low carbon emissions in the heat treatment industry. By highlighting the differences between aging methods, this study provides solutions for optimizing heat treatment processes and thereby achieving industrial advancement and sustainability goals.

## 1. Introduction

SUS 630 stainless steel, also known as 17-4 PH stainless steel, is a type of martensitic precipitation-hardening stainless steel [1]. Due to its superior mechanical properties and corrosion resistance, it is extensively used in high-performance applications such as the aerospace, petrochemical, and nuclear industries [2,3,4,5]. The optimization of SUS 630 stainless steel’s performance is primarily influenced by the conditions of the aging heat treatment [6,7,8]. By adjusting the aging temperature and duration, it is possible to control the formation and distribution of secondary precipitates, thereby altering the material’s mechanical properties and corrosion resistance [9,10].

Following solution treatment, SUS 630 stainless steel undergoes aging at various temperatures to promote precipitate formation. High-temperature aging facilitates the growth of larger precipitates, enhancing the steel’s hardness while reducing ductility [11]. In contrast, low-temperature aging encourages the formation of finer precipitates, achieving a balance between hardness and toughness [12]. Short aging duration generally lead to fine, dispersed precipitates, maintaining the steel’s toughness and ductility with only a modest increase in hardness and strength [13]. As the aging duration extends, the precipitates grow and cluster, significantly improving hardness and strength. However, excessively long aging duration can cause precipitate coarsening, which diminishes ductility and impact resistance [14]. Conducting studies on SUS 630 stainless steel with varying aging duration is essential for understanding the influences of precipitation behavior on the steel’s microstructure. This will ultimately assist in optimizing steel’s performance to meet the requirements for diverse applications.

The choice of aging treatment medium plays a significant role in the precipitation hardening and oxidation resistance of steel. Atmospheric aging treatment remains prevalent in the heat treatment industry due to its simplicity; however, exposure to oxygen at high temperatures causes the formation of a surface oxide layer, impacting surface hardness and corrosion resistance [15]. On the other hand, molten salt baths (e.g., sodium chloride, potassium chloride) offer reliable temperature uniformity and a controlled environment that prevents oxidation and promotes homogeneously distributed precipitates, enhancing the material’s hardness and durability [14,16]. Liquid metals, such as gallium, tin, and lead, due to their high thermal conductivity [17,18,19], enable rapid heating in shorter duration and effectively prevent oxidation, thereby improving the surface quality and fatigue resistance of steel.

Aging heat treatment conditions are a central factor for strengthening the properties of SUS 630 stainless steel [20]. This study also considers the use of environmentally conscious aging media to minimize energy consumption, carbon emissions, and environmental impact. Atmospheric aging treatment easily leads to surface oxidation of the steel and the generation of volatile emissions [21], while salt bath aging, though effective in preventing oxidation, may involve chemicals that pose environmental concerns [22]. Liquid metals, with their superior thermal conductivity, enable shorter heating duration and reduced energy consumption. Additionally, the recyclability of liquid metals minimizes waste and pollution. Based on this, this study employs liquid tin as an aging medium, aiming to reduce treatment duration and leverage its recyclability to lower the carbon footprint of the heat treatment process.

This study investigates the effects of aging treatment on USU 630 stainless steel with different temperatures, duration, and media (atmosphere and liquid tin), focusing on the relationships among microstructural evolution, hardness, strength, impact toughness, and fatigue resistance. The anticipated results are expected to provide valuable references for both academic research and industrial applications. Given the energy-intensive nature of the heat treatment required for SUS630 stainless steel, this study also evaluates the environmental impact of the aging process when using different media. Simultaneously, it introduces a sustainable approach to minimizing energy consumption and carbon emissions during the heat treatment process.

## 2. Materials and Methods

### 2.1. Materials

A cold-drawn SUS630 round bar (AJEI International Co., Ltd., Hsinchu, Taiwan) with a diameter of 5 mm was used in this study. The chemical composition of SUS630 stainless steel is presented in Table 1. The steel specimens were austenitized at 1050 °C for 30 min in an air furnace (DF20, Yu Kung Thermionic machinery Co., Ltd., Kaohsiung, Taiwan).

### 2.2. Methodology

The residual stress and retained austenite in the SUS630 specimens quenched in the air and water are summarized in Table 2. The differences in residual stress (compressive stress) between air-quenched and water-quenched specimens were minimal. Additionally, the traces of retained austenite detected indicated that most of the austenite had transformed into martensite. Based on these results, subsequent studies were conducted on the air-quenched specimens. The air-quenched steel specimens were aged using an air furnace at different temperatures (500–700 °C) for various duration (30–240 min) to evaluate the age-hardening behavior. For clarity, the SUS630 specimens were designated as Q1050, QA500, QA600, and QA700 according to their heat treatment conditions. In the tin bath heat treatment, 1000 g of lead-free pure tin (99.9%) was melted into a liquid state within a 314 stainless steel tank (Φ150 mm × 80 mm), serving as the aging medium.

### 2.3. Instrumental Methods

Microstructural analysis was performed using an optical microscope (OM, OLYMPUS BX51M, Tokyo, Japan). The surface characteristics and fracture surface of the aged specimens were analyzed using a scanning electron microscope (SEM, HITACHI S-3000N, Tokyo, Japan). Phase compositions of the age-hardened specimens were analyzed through X-ray diffraction spectroscopy (XRD, Bruker AXS, Karlsruhe, Germany) with a scanning of 2θ from 30° to 100°. Microhardness measurements were conducted using a microhardness tester (Mitutoyo, Kanagawa Japan), applying a 500 g load for 10 s. The hardness (HV) values were measured on the cross-section along the drawing direction of the round bar. Specimens for tensile, impact, and fatigue tests were machined from the stainless-steel round bars, with the dimensions shown in Figure 1. Each datum is the average of at least 3–5 specimens. Tensile tests were conducted using a universal testing machine (HT-2402, Hung Ta Instrument Corporation, Taichung, Taiwan) to evaluate the mechanical properties of steel aged in different media (atmosphere and liquid tin). The Charpy impact test was performed to determine the impact toughness of the aged specimens, employing a pendulum with a mass of 29.07 kg, a length of 0.75 m, and an initial angle of 140°. Dynamic fatigue tests were conducted using a fatigue testing machine (HT-9711, Hung Ta Instrument Corporation, Taichung, Taiwan), applying alternating stress (±40 kN) at a frequency of 5 Hz to assess the fatigue resistance of the aged specimens.

## 3. Results

The microstructural characteristics of specimens age-hardened in an air furnace are shown in Figure 2. After atmospheric quenching (Figure 2a), the microstructure of the Q1050 specimen predominantly consists of lath martensite [23]. In the age-treated steel matrix, quenched martensite (appearing as black regions) and tempered martensite (grey regions) grow along the orientation of the austenite grain. Notably, a considerable distribution of fine particles is observed within the lath martensite structures, which may be associated with the formation of M_7_C_3_ and NbC carbides [24]. When the aging temperature increases to 600 °C (Figure 2b), the microstructure of the steel exhibits an obvious transformation. The crystal size increases with an increase in aging temperature, indicative of over-aging precipitation behavior [25]. Additionally, higher aging temperatures accelerate the conversion of quenched martensite to tempered martensite, resulting in an increase in tempered martensite [26]. Figure 3 displays SEM images of 630 stainless steel specimens aged at various temperatures (500–600 °C) in an atmospheric furnace. The lath martensite structures progressively become finer and more uniform with an increase in aging temperature. At elevated aging temperatures, the grain boundaries of the stainless steel become more prominent, and the martensite structures are enclosed within them. Furthermore, fine particles are abundantly distributed along the grain boundaries and within the grains, indicating the formation of the precipitation hardening phase. As the aging temperature rises, both the quantity and size of precipitated phases increase. Briefly, precipitates located at the grain boundaries enhance boundary strength, while those within the martensite structure restrict dislocation movement, thereby improving the strength of the 630 stainless steel.

Figure 4 displays the phase compositions of 630 steel specimens subjected to various aging treatments. All specimens present distinct diffraction peaks at (110), (200), (211), and (211), corresponding to the α’-martensite phase [27]. Notably, a weak diffraction peak appears adjacent to the (110) α’ phase, indicating the presence of a retained austenite ($\gamma_{Ri}$) phase in the (111) orientation. This result aligns with the retained austenite detected in the SUS630 steel (Table 2). The primary phases in the aged specimens are martensite and a minor amount of retained austenite. Importantly, no precipitation phases are observed in the XRD patterns, suggesting that the precipitation phase content is below the detection sensitivity. As the aging temperature rises, the intensity of the (110) diffraction peak increases, indicating alterations in the crystallization characteristics of martensite. Additionally, the full width at half maximum (FWHM) of the α’ (110) diffraction peak broadens (0.39 to 0.49) with higher aging temperatures (600–700 °C), reflecting an increase in martensite crystal size at elevated temperatures [28]. This finding is consistent with the microstructural observations in Figure 2, reinforcing the idea that higher aging temperatures significantly impact the mechanical properties of 630 stainless steel.

This study explored the effects of aging treatment conditions (temperature, duration, and medium) on the precipitation-hardening behavior of 630 steel specimens, with the results illustrated in Figure 5. The hardness value of the solution-treated specimen (Q1050) was approximately HV 370 (Figure 5a). Aging in an atmospheric furnace revealed that 630 stainless steel achieved its maximum hardness value after being aged at 500 °C for 1 h. This increase in hardness is attributed to the precipitation of nanoscale Cu-rich phases within the martensite matrix, which enhances the material’s hardness. However, aging at higher temperatures (600 and 700 °C) for 1 h caused a significant decrease in hardness, suggesting that higher temperatures accelerate precipitation behavior but also promote the coarsening of the secondary precipitates. The influence of aging duration on hardness was assessed by subjecting 630 stainless steel to aging at 500 °C for varying duration (Figure 5b). The maximum hardness was observed after an aging duration of 60 min, primarily due to the formation of nanoscale strengthening phases during the aging process. However, further increases in aging duration did not significantly impact the precipitation hardening of 630 stainless steel. This can be attributed to the low degree of carbon saturation in the martensite matrix in the steel, resulting in minimal tempering effects on hardness [20]. Therefore, in this study, the carbon content of the given 630 stainless steel was below 0.07 wt.%, which influences its hardening behavior during the aging process. To evaluate the effectiveness of using liquid tin as an aging medium, the hardening behavior of 630 stainless steel subjected to Sn bath aging is presented in Figure 5c. The 630 stainless steel specimens aged in the Sn bath for different duration exhibited minor variations in hardness, with a slight improvement observed after 60 min. However, hardness values for the specimens aged in the liquid Sn were notably lower than those aged in an atmospheric furnace. This reduction in hardness is likely due to the potential dissolution of alloying elements into the liquid Sn, which may diminish the formation of strengthening phases and thus decrease the effectiveness of precipitation hardening [29]. Although the choice of aging medium influenced surface hardness, the influence on their tensile strength remains unclear. To address this, tensile tests was conducted on the steel specimens subjected to different aging treatments.

The deformation resistance of 630 stainless steel specimens at different atmospheric aging temperatures was assessed through tensile testing, as illustrated in Figure 6. Stress–strain curves derived from tensile tests performed on steel specimens aged at different temperatures are presented in Figure 6a. Notably, the QA500 specimen revealed a considerable area under the stress–strain curve, indicating high deformation resistance, a characteristic of strengthened steel. With an increase in aging temperature, the tensile strength of the steel specimen decreased substantially, while its ductility improved, as seen in Figure 6b. All steel specimens exhibited a total elongation of approximately 20% after aging heat treatment, indicating stable plastic deformation behavior (Figure 6c). These results reveal that 630 stainless steel aged at 500 °C demonstrates effective precipitation hardening and exhibits high toughness with uniform deformation. Based on these tensile test results, additional aging treatments were conducted on the steel specimens using 500 °C molten Sn to further examine the influence of the Sn bath medium on the precipitation hardening characteristics of 630 stainless steel.

The tensile properties of the 630 stainless steel obtained after aging in a 500 °C Sn bath for various duration (30–90 min) are shown in Figure 7. The tensile strength of the steel specimens remained relatively constant across different Sn bath duration. Notably, the steel specimens acquired significant precipitation hardening even with a short Sn bath aging duration of 30 min, exhibiting a total elongation exceeding 25%. Compared to the atmosphere-aged specimen (QA500), the specimen aged in a Sn bath for 30 min demonstrated comparable tensile strength (~1300 MPa), indicating that the Sn bath heat treatment efficiently promotes age hardening. The heat capacity of liquid Sn at 500 °C (Cp = 6.78 cal/mol·°K) [30] and its thermal conductivity (>30 W/m·°K) compared to the atmosphere (~0.024 W/m·°K) [31] make it an effective medium for metal tempering and aging. This high thermal conductivity allows liquid Sn to significantly reduce heat treatment time and energy consumption.

Figure 8 displays the impact toughness of 630 stainless steel specimens with varying aging duration under atmospheric and Sn bath aging conditions. Compared to the atmosphere-aged specimens, those aged in a Sn bath exhibit significantly higher impact toughness. This enhancement can be attributed to the stable heating environment provided by the Sn bath during the precipitation hardening process. Due to the high thermal conductivity of liquid Sn, the Sn bath can effectively control the morphology and distribution of precipitates [32], improving the toughness of the 630 stainless steel specimens. In addition, the liquid Sn maintains chemical stability at elevated temperatures [33], which reduces with the introduction of the steel during heat treatment. In contrast to atmospheric aging, Sn bath aging prevents oxidization by isolating the metal from oxygen, thereby preserving the surface quality. Note that the impact toughness of Sn bath-aged specimens fluctuates significantly, potentially due to incomplete immersion in the liquid Sn during the aging process. Post-impact fractography (Figure 9) reveals that all 630 steel specimens exhibit ductile fracture with dimple structures. The atmosphere-aged specimens aged for 30 min show a combination of micron- and nanoscale dimples (Figure 9a). As aging duration increases, the density of the micron-scale dimples decreases, shifting to nanoscale dimples, which correlates with higher impact energy (Figure 9b). In Sn bath-aged specimens, dimple structures are uniformly distributed across the fracture surface. Specimens aged for extended periods in the Sn bath show coarser dimples than those aged for 30 min (Figure 9c), leading to lower impact energy. However, the dimple sizes in Sn bath-aged specimens are more uniform than in atmosphere-aged specimens, highlighting the role of liquid Sn’s high thermal conductivity in refining and uniformly distributing precipitates, thereby improving impact toughness.

Figure 10 illustrates the fatigue limits of 630 stainless steel specimens aged in an atmospheric furnace and a Sn bath, as measured by dynamic tensile testing. The atmosphere-aged specimen, treated for 60 min, demonstrates a high fatigue resistance, in line with its notable tensile strength and hardness. Notably, Sn bath aging for just 30 min achieves fatigue limits comparable to those of specimens aged for 60 min in the atmosphere. This suggests that the Sn bath aging process effectively enhances fatigue resistance in a shorter time frame. Under extreme conditions in dynamic fatigue tests, the fatigue resistance of 630 stainless steel correlates closely with its tensile strength, ductility, and hardness. High tensile strength helps resist crack propagation under cyclic loading, while high ductility enables plastic deformation post-crack initiation, reducing local stress concentration and slowing crack propagation. The ductility of the Sn bath-aged steel is a key factor contributing to its extended dynamic fatigue life. Furthermore, the secondary phases formed during the aging process were uniformly distributed throughout the ductile matrix, thereby effectively enhancing resistance to dynamic fatigue. These finely dispersed phases restrict dislocation movement and impede crack propagation, contributing to the overall durability and performance of the 630 stainless steel specimens under cyclic loading conditions.

In accordance with the above results, Table 3 compares the mechanical properties and environment benefits of atmosphere- and Sn bath-aged 630 stainless steel, underscoring their contributions to energy conservation and emission reduction. The required mechanical properties and hardness of the given 630 stainless steel were primarily achieved through solution-aging treatment. Regardless of whether the steel was aged in the atmosphere or in the Sn bath, it exhibited excellent mechanical properties. Notably, aging in liquid Sn significantly reduced the required aging duration. In terms of carbon emissions, energy conservation, and economic efficiency, the benefits of 30 min of Sn bath aging were compared with those of 60 min of atmospheric aging. For the 500 °C aging treatment, the box-type electric furnace used in present study operates with an average power consumption of 2000 W. Assuming an 8 h workday and a 5-day workweek, the total annual working time is 1920 h. For applying the age heat treatment to workpieces of the same weight, by reducing processing duration by 50% with Sn bath aging, the estimated energy consumption drops to 1920 kWh. The carbon emissions of the Sn bath aging were calculated using the 2023 electricity emission factor, as published by the Energy Administration, Ministry of Economic Affairs, Taiwan.


CO_2_ emission = Power Consumption (kWh) × 0.494 (kg CO_2_e/kWh)(1)


Based on the above calculation formula, this results in 0.9485 ton of CO_2_ emissions annually for the Sn bath aging treatment. In addition, the energy consumption required to heat a 1 kg Sn ingot to 500 °C should also be calculated to evaluate the CO_2_ emissions of the process. The heat requirements for heating 1 kg of solder balls from 25 to 500 °C can be divided into three distinct components:
*Q*_total_ = *Q*_1_ + *Q*_2_ + *Q*_3_(2)
(3)$$Q_{1}=m\times \;c\times \;{∆T}_{1}$$
*Q*_2_ = *m* × *L*_f_(4)
(5)$$Q_{3}=m\times \;c\times \;{∆T}_{2}$$
where:
*Q*_1_ = Heating the solid tin from 25 °C to its melting point (231.9 °C)*Q*_2_ = Melting the tin at 231.9 °C*Q*_3_ = Heating the liquid tin from 231.9 to 500 °C*m* = 1 kg (mass of tin)*c* = 0.227 kJ/kg·°C (specific heat capacity of tin)∆*T*_1_ = Temperature change from 25 to 231.9 °C*L_f_* = 59.2 kJ/kg (latent heat of fusion of tin)∆*T*_2_ = Temperature change from 231.9 to 500 °C

The total heat required to transition the solid tin into liquid tin at 500 °C is approximately 167 kJ. By converting heat into electricity consumption (1 kWh = 3600 kJ) and accounting for the heating efficiency of the electric furnace (80%), the actual energy consumption is estimated to be approximately 0.058 kWh. Therefore, the use of liquid tin as the heating medium during the aging treatment stage resulted in CO_2_ emissions of nearly 1 ton. The cost of electricity consumed for the aging treatment is calculated by multiplying the total power consumption (in kWh) by the unit electricity rate. In summary, Sn bath aging of 630 stainless steel offers energy efficiency, cost savings, and reduced environmental impact while delivering robust precipitation strengthening. Due to its physical properties and recyclability, liquid Sn presents a viable low-carbon alternative for heat treatment processes in the industry.

## 4. Conclusions

(1)In the case of SUS 630 stainless steel, heat treatment induces a martensitic phase transformation, while the aging process promotes the formation of secondary precipitates, both of which are critical to enhancing the steel’s strength and hardness. The present study introduces liquid Sn metal as a medium for aging treatment to explore both the strengthening effects of Sn bath aging on 630 stainless steel and its contribution to environmental sustainability.(2)The aging process of 630 stainless steel promotes a dense distribution of carbides and precipitates within the martensite matrix, which are the primary contributors to the steel’s performance.(3)Following aging at 500 °C for 60 min in an atmospheric furnace, the 630 stainless steel demonstrates superior precipitation hardening and reliable ductility (>20%). When aged in a 500 °C Sn bath, the 630 stainless steel achieves similar mechanical enhancements in half the duration required by atmospheric aging, displaying improved fatigue limits due to its high tensile strength and ductility.(4)Additionally, incorporating liquid Sn as the aging medium significantly increases processing efficiency and contributes to environmental sustainability by reducing the time and energy required for heat treatment. This approach highlights Sn bath aging as a promising, low-emission alternative for the heat treatment industry, offering an efficient route to achieving both enhanced material properties and lower environmental impact.(5)The liquid Sn may chemically react with certain high-alloy steels or coatings, limiting its applications. Additionally, as a rare metal, tin is relatively expensive, resulting in higher maintenance costs for heat treatment, especially in large-scale applications.

## Figures and Tables

**Figure 1 materials-18-00574-f001:**
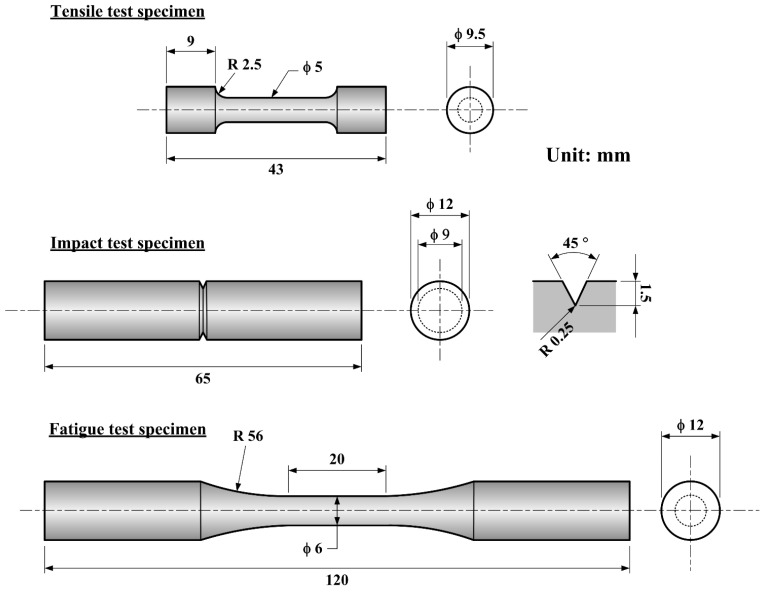
Dimensions of tensile, impact and fatigue test specimens.

**Figure 2 materials-18-00574-f002:**
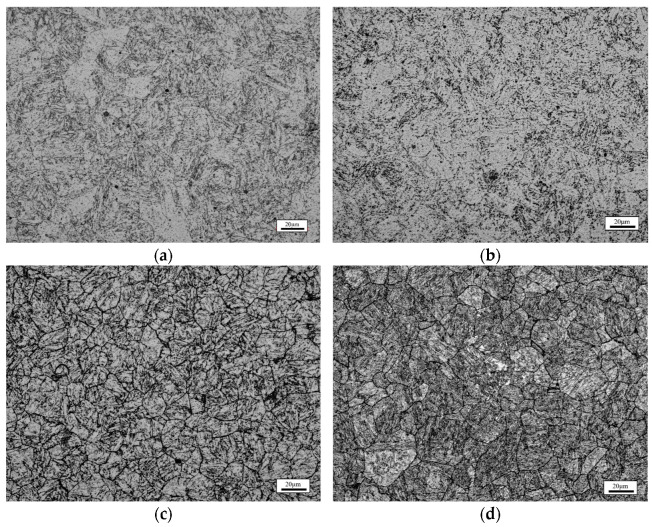
Microstructure characteristics of the (**a**) Q1050, (**b**) QA500, (**c**) QA600, and (**d**) QA700 specimens.

**Figure 3 materials-18-00574-f003:**
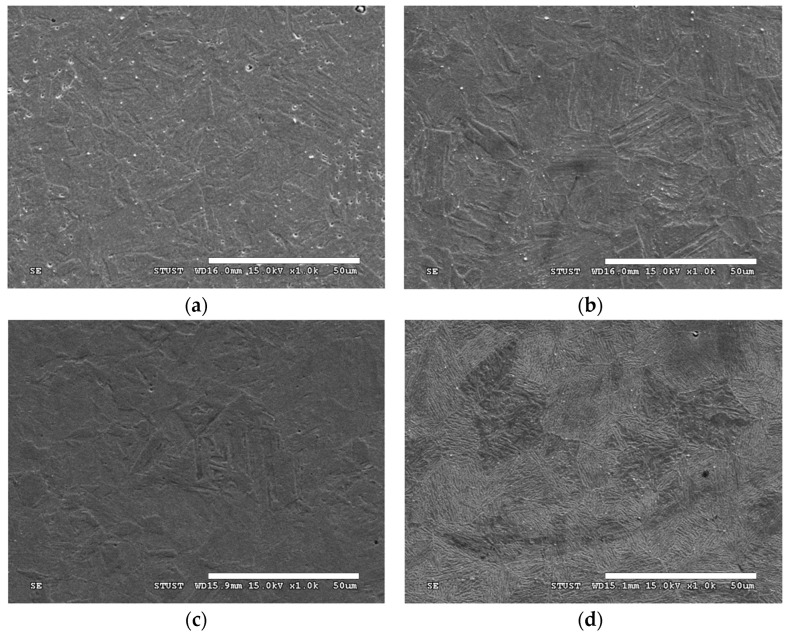
SEM images of the (**a**) Q1050, (**b**) QA500, (**c**) QA600, and (**d**) QA700 specimens.

**Figure 4 materials-18-00574-f004:**
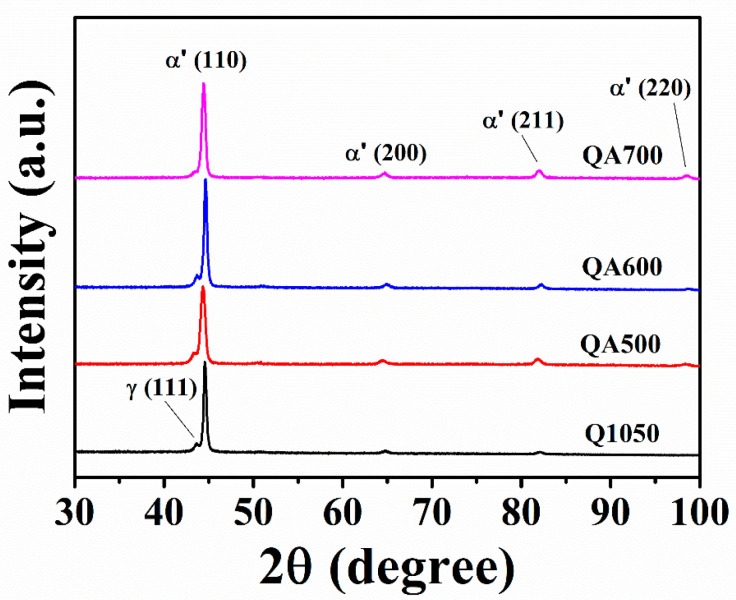
XRD patterns of the SUS 630 stainless steel specimens obtained after solution-aging treatments.

**Figure 5 materials-18-00574-f005:**
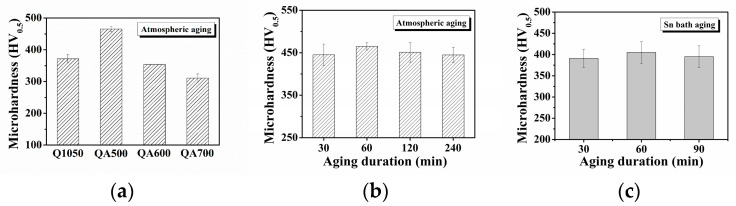
The variation in microhardness of 630 stainless steel obtained under three conditions: (**a**) atmospheric aging at different temperatures, (**b**) aging at 500 °C in an atmospheric environment for varying duration, and (**c**) aging at 500 °C in a Sn bath for varying duration.

**Figure 6 materials-18-00574-f006:**
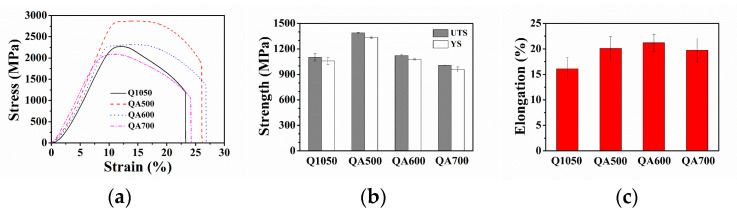
Tensile properties of 630 stainless steel aged at different temperatures: (**a**) stress–strain curve, (**b**) tensile strength, and (**c**) elongation.

**Figure 7 materials-18-00574-f007:**
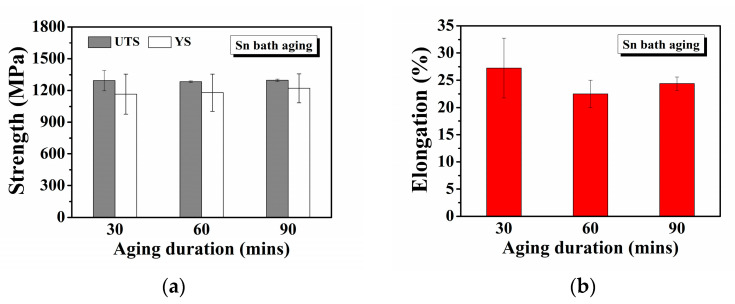
Tensile properties of 630 stainless steel with different aging temperatures: (**a**) tensile strength, and (**b**) elongation.

**Figure 8 materials-18-00574-f008:**
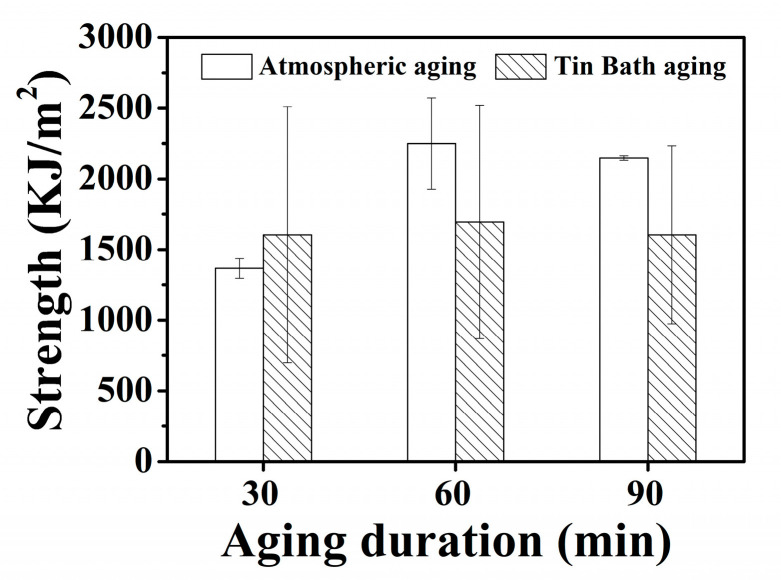
Impact toughness as a function of aging duration for 630 steel specimens in atmosphere and liquid Sn environments.

**Figure 9 materials-18-00574-f009:**
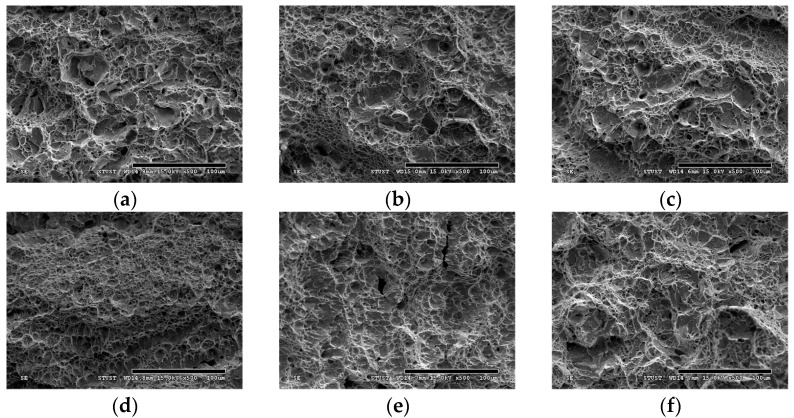
Fracture surfaces of 630 stainless steel specimens with varying aging duration following impact test: atmospheric aging for (**a**) 30 min, (**b**) 60 min, and (**c**) 90 min; Sn bath aging for (**d**) 30 min, (**e**) 60 min, and (**f**) 90 min.

**Figure 10 materials-18-00574-f010:**
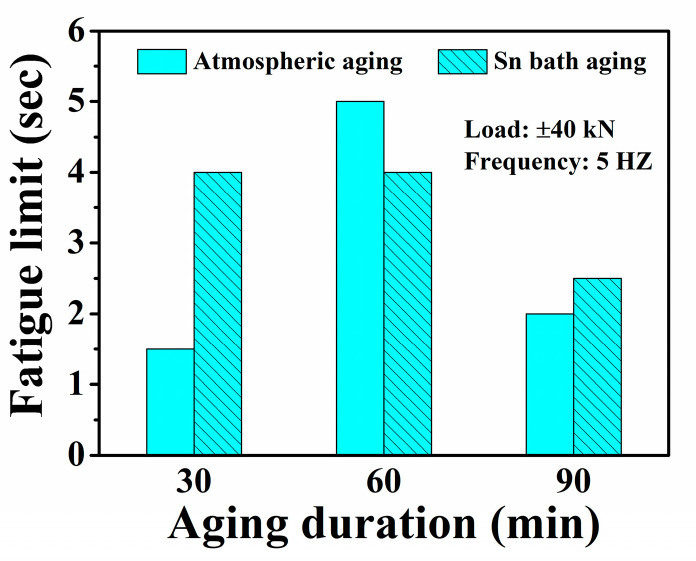
Fatigue limit as a function of aging duration for the 630 stainless steel specimens in atmosphere and liquid Sn environments.

**Table 1 materials-18-00574-t001:** Chemical composition of SUS630 round bar.

Wt.%	C	Si	Mn	P	S	Ni	Cr	Cu	Nb
SUS630	<0.07	<1.00	<1.00	<0.04	<0.03	3.00–5.00	15.00–17.50	3.00–5.00	0.15–0.45

**Table 2 materials-18-00574-t002:** Measurements of residual stress and retained austenite of SUS 630 after quenching.

Quenching Medium	Residual Stress (MPa)	Retained Austenite (%)
Air	−365	2.3
Water	−437	1.1

**Table 3 materials-18-00574-t003:** The mechanical properties and environment benefits of 630 stainless steel obtained after being aged in a Sn bath and an atmospheric furnace.

		30 min Sn Bath Aging	60 min Atmospheric Aging
Mechanical properties	Tensile strength	○	◎
	Elongation	◎	○
	Hardness	○	◎
	Fatigue limits	○	○
Benefits	Energy consumption (kWh)	1920	3840
	Carbon emissions (tCO_2_e/year)	0.9772	1.8970
	Economics (TWD)	8256	16,512

Note: **◎** Superior, **○** Premium. Assume the average cost of industrial electricity is TWD 4.3 per kWh.

## Data Availability

The original contributions presented in this study are included in the article. Further inquiries can be directed to the corresponding author.

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
