# Peer review of "Microstructures and Mechanical Properties of SUS 630 Stainless Steel: Effects of Age Hardening in a Tin Bath and Atmospheric Environments"

_materials, 2025, doi:10.3390/ma18030574_

Round 1

Reviewer 1 Report

Comments and Suggestions for Authors

This study introduces a potential heat treatment method for SUS 630 stainless steel as an alternative to traditional atmospheric aging. The paper presents metallographic and mechanical experiments to evaluate the effectiveness of this new approach. The analysis and logic are robust, and the work can be considered for publication following minor revisions:

1.      The model of the SEM should be specified in the experimental section, as currently, only the optical microscope is mentioned.

2.      The scale bars in the morphological figures should be consistent. For instance, the scale bar in Figure 3 is less clear compared to Figure 2, where the added legend is more prominent.

3.      The full width at half maximum (FWHM) of the peaks in Figure 4 is not clearly perceivable, making it difficult to support the conclusion that "The FWHM of the (110) diffraction peak broadens with higher aging temperatures." The authors might consider zooming in on Figure 4 or providing the FWHM values in a table or text for clarity.

4.      The caption of Figure 5 is wrong.

5.      The number of samples used for the mechanical tests should be specified, either in the experimental section or in the figure captions.

6.      Figure 9 appears to be SEM images; however, this is not specified in the caption or the text.

7.      The authors suggest that incomplete immersion may have caused the fluctuation in impact toughness. Providing a more detailed explanation of the immersion procedure would enhance clarity. For example, what is the volume of the tin tank, how deep was the immersion, and would a greater volume of tin improve the impact toughness?

8.      The amount of tin used for immersion should influence energy consumption, given the constant heat capacity of liquid tin. This relates to the energy estimation in the final section of the manuscript. The authors are encouraged to elaborate on the energy calculations, such as specifying the model of the heating system used for the tin bath and the quantity of tin considered in these calculations.

Author Response

Reviewer 1

This study introduces a potential heat treatment method for SUS 630 stainless steel as an alternative to traditional atmospheric aging. The paper presents metallographic and mechanical experiments to evaluate the effectiveness of this new approach. The analysis and logic are robust, and the work can be considered for publication following minor revisions:

  1. The model of the SEM should be specified in the experimental section, as currently, only the optical microscope is mentioned.

Response:

The model of the SEM (HITACHI S-3000N, Tokyo, Japan) has been added in the experimental section.

  1. The scale bars in the morphological figures should be consistent. For instance, the scale bar in Figure 3 is less clear compared to Figure 2, where the added legend is more prominent.

Response:

The scale bar in Figure 3 has been modified in the revision paper.

  1. The full width at half maximum (FWHM) of the peaks in Figure 4 is not clearly perceivable, making it difficult to support the conclusion that "The FWHM of the (110) diffraction peak broadens with higher aging temperatures." The authors might consider zooming in on Figure 4 or providing the FWHM values in a table or text for clarity.

Response:

From the observation of Fig. 2, the microstructure of the aged steel undergoes a notable transformation as the aging temperature rises to 600 °C. Therefore, the FWHM of the α’ (110) diffraction peak broadens from 0.39 to 0.49 when the aging temperature increases from 600 °C to 700 °C. The FWHM values of the (110) diffraction peak for the aged steel specimens have been incorporated into the revision manuscript.

  1. The caption of Figure 5 is wrong.

Response:

Thanks to reviewer’s comments. The caption of Figure 5 has been revised to “The variation in microhardness of 630 stainless steel obtained under three conditions: (a) atmospheric aging at different temperatures, (b) aging at 500 °C in an atmospheric environment for varying durations, and (c) aging at 500 °C in a Sn-bath for varying durations.” in the revision paper.

  1. The number of samples used for the mechanical tests should be specified, either in the experimental section or in the figure captions.

Response:

The number of samples used for the mechanical tests have been specified in the experimental section. This description ”Each datum is the average of at least 3~5 specimens.” has been added in the revision paper.

  1. Figure 9 appears to be SEM images; however, this is not specified in the caption or the text.

Response:

The fracture surface of 630 stainless steel specimen with varying aging durations were examined through a scanning electron microscope. This description has been revised into in the experimental section.

  1. The authors suggest that incomplete immersion may have caused the fluctuation in impact toughness. Providing a more detailed explanation of the immersion procedure would enhance clarity. For example, what is the volume of the tin tank, how deep was the immersion, and would a greater volume of tin improve the impact toughness?

Response:

Based on reviewer’s comments, this description “In the tin bath heat treatment process, 1000 g of lead-free pure tin (99.9%) is melted into a liquid state within a 304 stainless steel tank (Φ150 mm´80mm), serving as the aging treatment medium.” has been added in the experimental section. In addition, based on the volume of tin tank, a given test specimens would be completely immersed in liquid tin. However, the 630 stainless steel has similar density characteristics to a liquid tin. Therefore, in liquid tin bath, a specially designed fixture mechanism can be implemented to ensure uniform heat transfer and minimize the effects of buoyancy interference.

  1. The amount of tin used for immersion should influence energy consumption, given the constant heat capacity of liquid tin. This relates to the energy estimation in the final section of the manuscript. The authors are encouraged to elaborate on the energy calculations, such as specifying the model of the heating system used for the tin bath and the quantity of tin considered in these calculations.

Response:

According to the reviewer’s comments, the calculation of energy consumption and carbon emissions required to heat 1 kg of tin ingot to 500 °C have been added in this revision paper (from line 318 to 340).

In addition, the energy consumption required to heat 1 kg-Sn ingot to 500 °C should also be calculated to evaluate the CO2 emission of process. The heat requirements for heating 1 kg of solder balls from 25 to 500°C can be divided into three distinct components:

                     Qtotal= Q1+Q2+Q3                               (1)

                     Q1= m ´ c ´                                 (2)

                     Q2= m ´ Lf                                     (3)

                     Q3= m ´ c ´                                 (4)

where:

Q1= Heating the solid tin from 25 °C to its melting point (231.9 °C)

Q2= Melting the tin at 231.9 °C

Q3= Heating the liquid tin from 231.9 to 500 °C

m= 1 kg (mass of tin)

c= 0.227 kJ/kg·°C (specific heat capacity of tin)

∆T1= Temperature change from 25 to 231.9 °C

Lf= 59.2 kJ/kg (latent heat of fusion of tin)

∆T2= Temperature change from 231.9 to 500 °C

The total heat requited to transition the solid tin into a liquid tin at 500 °C is approximately 167 kJ. By converting heat into electricity consumption (1 kWh= 3600 kJ) and accounting for the heating efficiency of the electric furnace (80 %), the actual energy consumption is estimated to be approximately 0.058 kWh. Therefore, the used of liquid tin as the heating medium during the aging treatment stage resulted in the CO2 emission of nearly 1 ton.

Reviewer 2 Report

Comments and Suggestions for Authors

The subject of the research and scientific novelty: the manuscript “ Microstructures and mechanical properties of SUS 630 stainless steel: Effects of age hardening in a tin−bath and atmospheric environments” investigates the solution-aging treatment of precipitation-hardening SUS 630 stainless steel, alongside an analysis of carbon emission generated by the energy consumption during aging treatments. By employing atmospheric and liquid tin as aging media, the research com prehensively explores the effects of aging treatments on the characteristics of 630 stainless steel. The maximum hardness value for the 630 stainless steel was observed after atmospheric aging at 500 C for 1 hour. The given 630 stainless steel obtained its maximum hardness value after atmospheric aging at 500 C for 1 h, indicating that the formation of secondary precipitates strengthens the steel’s performance. The Sn−bath treatment, recognized in metallurgical science and heat treatment for its excellent thermal conductivity and recyclability, underscores its potential to enhance process efficiency and support low-carbon emissions in the heat  treatment industry.

General comments: The work is thematically appropriate for the Materials journal. The concept is original and novel. There are several issues with the manuscript's preparation which led to the 'major review' decision. Text is written in manner that is undestandable. Introduction provides a good state of the art. The proposed hypotheisis is testable and repeatable, and the authors managed to prove it. Results are presented anddiscussed in neat and understandable manner. Most of the figures and tables are of good quality. Conclusions have to be reformatted.  Explanations and suggestions are in Specific comments.

Specific comments:

  1. The manuscript’s title is explanatory. It adequately describes the conducted study.
  2. Key words are adequate.
  3. The abstract is informative and of adequate length. The specific results are provided. Methodology is explained. Please add the scientific novelty/gap in knowledge in the last sentence.
  4. The Introduction section gives an adequate preview of the main idea behind the work. State of the art review is adequately conducted. The goals of the study are mentioned. A line or two about the scientific contribution of this paper to global knowledge would be good.
  5. Experimental chapter; Please divide the text into subchapters: (1) materials; (2) methodology; (3) instrumental methods.
  6. Results are adequately presented and discussed. The discussion is logical and follows the proposed thesis.
  7. All of the figures and tables are necessary. The presented data is adequate. Table 3 could be smaller.
  8. Please provide the main conclusions as list or bullets. Mention the limitations of the study and future investigations.
  9. Literature is adequate and up to date.
  10. The English language is understandable. There are some formatting errors.  

Author Response

Reviewer 2

The subject of the research and scientific novelty: the manuscript “ Microstructures and mechanical properties of SUS 630 stainless steel: Effects of age hardening in a tin−bath and atmospheric environments” investigates the solution-aging treatment of precipitation-hardening SUS 630 stainless steel, alongside an analysis of carbon emission generated by the energy consumption during aging treatments. By employing atmospheric and liquid tin as aging media, the research com prehensively explores the effects of aging treatments on the characteristics of 630 stainless steel. The maximum hardness value for the 630 stainless steel was observed after atmospheric aging at 500 °C for 1 hour. The given 630 stainless steel obtained its maximum hardness value after atmospheric aging at 500 °C for 1 h, indicating that the formation of secondary precipitates strengthens the steel’s performance. The Sn−bath treatment, recognized in metallurgical science and heat treatment for its excellent thermal conductivity and recyclability, underscores its potential to enhance process efficiency and support low-carbon emissions in the heat  treatment industry.

General comments: The work is thematically appropriate for the Materials journal. The concept is original and novel. There are several issues with the manuscript's preparation which led to the 'major review' decision. Text is written in manner that is undestandable. Introduction provides a good state of the art. The proposed hypotheisis is testable and repeatable, and the authors managed to prove it. Results are presented anddiscussed in neat and understandable manner. Most of the figures and tables are of good quality. Conclusions have to be reformatted.  Explanations and suggestions are in Specific comments.

  1. The manuscript’s title is explanatory. It adequately describes the conducted study.

Response:

We appreciate reviewer’s valuable comments on this revision manuscript.

  1. Key words are adequate.

Response:

We appreciate reviewer’s valuable comments on this revision manuscript.

  1. The abstract is informative and of adequate length. The specific results are provided. Methodology is explained. Please add the scientific novelty/gap in knowledge in the last sentence.

Response:

We appreciate reviewer’s valuable comments on this revision manuscript. This sentence “By comparing the differences in aging methods, this study provides solutions for optimizing heat treatment processes, thereby achieving industrial advancement and sustainability goals.” has been added in this revision paper.

  1. The Introduction section gives an adequate preview of the main idea behind the work. State of the art review is adequately conducted. The goals of the study are mentioned. A line or two about the scientific contribution of this paper to global knowledge would be good.

Response:

Thanks reviewer’s valuable comments on this revision manuscript. According to reviewer’s comments, this sentence “Simultaneously, it introduces a sustainable approach to minimizing energy consumption and carbon emissions during the heat treatment process.” has been added in the introduction section.

  1. Experimental chapter; Please divide the text into subchapters: (1) materials; (2) methodology; (3) instrumental methods.

Response:

Thanks reviewer’s valuable comments on this revision manuscript. The subchapters “2.1 Materials, 2.2 Methodology, 2.3 Instrumental methods” have been added in the revision text.

  1. Results are adequately presented and discussed. The discussion is logical and follows the proposed thesis.

Response:

We appreciate reviewer’s positive comments on this article.

  1. All of the figures and tables are necessary. The presented data is adequate. Table 3 could be smaller.

Response:

The format of Table 3 has been revised in the revision manuscript.

  1. Please provide the main conclusions as list or bullets. Mention the limitations of the study and future investigations.

Response:

The conclusion has been revised to a list format in this manuscript. In addition, the limitations of Sn-bath treatment have also been added to the revision paper, as follow:

  • The liquid Sn may chemically react with certain high alloy steels or coatings, limiting its applications. Additionally, as a rare metal, tin is relatively expensive, resulting in higher maintenance costs for heat treatment, especially in large-scale applications.

  1. Literature is adequate and up to date.

Response:

We appreciate reviewer’s positive comments on this article.

  1. The English language is understandable. There are some formatting errors.

Response:

Throughout this manuscript, the English language and formatting errors have been modified.

Round 2

Reviewer 2 Report

Comments and Suggestions for Authors

The Authors have answered all of the questions and accepted the suggestions given in the review. The manuscript has been sufficiently improved to warrant publication in Materials.